# Molecular Mechanisms Underlying the Retrieval and Extinction of Morphine Withdrawal-Associated Memories in the Basolateral Amygdala and Dentate Gyrus

**DOI:** 10.3390/biomedicines10030588

**Published:** 2022-03-02

**Authors:** Aurelio Franco-García, Francisco José Fernández-Gómez, Victoria Gómez-Murcia, Juana M. Hidalgo, M. Victoria Milanés, Cristina Núñez

**Affiliations:** 1Group of Cellular and Molecular Pharmacology, Department of Pharmacology, CEIR Campus Mare Nostrum, University of Murcia, 30120 Murcia, Spain; aurelio.franco@um.es (A.F.-G.); franciscojose.fernandez@um.es (F.J.F.-G.); victoria.gomez2@um.es (V.G.-M.); jmhc@um.es (J.M.H.); 2Instituto Murciano de Investigación Biosanitaria (IMIB)—Arrixaca, 30120 Murcia, Spain

**Keywords:** morphine withdrawal, conditioned place aversion, memory, dentate gyrus, basolateral amygdala, mTOR, Arc, Homer1, NMDA receptors

## Abstract

Despite their indisputable efficacy for pain management, opiate prescriptions remain highly controversial partially due to their elevated addictive potential. Relapse in drug use is one of the principal problems for addiction treatment, with drug-associated memories being among its main triggers. Consequently, the extinction of these memories has been proposed as a useful therapeutic tool. Hence, by using the conditioned place aversion (CPA) paradigm in rats, we investigated some of the molecular mechanisms that occurr during the retrieval and extinction of morphine withdrawal memories in the basolateral amygdala (BLA) and the hippocampal dentate gyrus (DG), which control emotional and episodic memories, respectively. The retrieval of aversive memories associated with the abstinence syndrome paralleled with decreased mTOR activity and increased Arc and GluN1 expressions in the DG. Additionally, Arc mRNA levels in this nucleus very strongly correlated with the CPA score exhibited by the opiate-treated rats. On the other hand, despite the unaltered mTOR phosphorylation, Arc levels augmented in the BLA. After the extinction test, Arc and GluN1 expressions were raised in both the DG and BLA of the control and morphine-treated animals. Remarkably, Homer1 expression in both areas correlated almost perfectly with the extinction showed by morphine-dependent animals. Moreover, Arc expression in the DG correlated strongly with the extinction of the CPA manifested by the group treated with the opiate. Finally, our results support the coordinated activity of some of these neuroplastic proteins for the extinction of morphine withdrawal memories in a regional-dependent manner. Present data provide evidence of differential expression and activity of synaptic molecules during the retrieval and extinction of aversive memories of opiate withdrawal in the amygdalar and hippocampal regions that will likely permit the development of therapeutic strategies able to minimize relapses induced by morphine withdrawal-associated aversive memories.

## 1. Introduction

Opioids are the most effective drugs used for the management of moderate to high intensity acute and chronic pain. Nonetheless, their propensity to induce misuse and abuse exerts an important deterrent effect for their prescription [1]. In fact, misuse of, and addiction to, prescription analgesic opioids was at the origin of the United States opioids crisis and, as a consequence, an increment in consumption of, first, synthetic analgesic opioids and, lately, illegal opioids such as heroin and illicit fentanyl, occurred [2]. An opioid crisis is also present in Europe, where it has become a progressively harmful public health problem [3].

Drug addiction is a brain condition defined by compulsive drug intake despite its adverse consequences and by frequent relapses in drug use, which limit the success of its treatment [4,5]. Among the factors that lead to the maintenance and reinstatement of drug seeking behaviours are the environmental stimuli associated with the substance of abuse [6], which, after repeated associations, gain secondary motivational value and are able to abnormally activate the learning and memory brain systems, thus promoting drug-directed behaviours over those guided by natural stimuli. The motivational value of drug-associated cues can last very long periods of time, even years [7]. Therefore, in the last decade, therapeutic approaches based on the extinction of drug-associated memories has been proposed for preventing drug seeking and relapse [4,5].

Abundant research has been performed to disentangle the mechanisms underlying the extinction of drug rewarding memories [8,9,10]. Nonetheless, less is yet known about the extinction of aversive memories associated with drug withdrawal [11,12]. In the conditioned place aversion (CPA) paradigm, which has been considered as a measure for drug-seeking behaviour, the negative symptoms of drug withdrawal are paired with a particular environment that consequently acquires long-lasting secondary motivational value and that afterwards, when tested, animals avoid [13]. One procedure to achieve the extinction of drug-directed behaviours consists in the ‘reconditioning’ of the animals by the administration of the drug’s vehicle before being confined in both the neutral and the previously withdrawal-associated contexts [7,11,13,14]. Extinction implies a diminution or inhibition of the conditioned response to the drug, but neither ‘unlearning’ nor ‘forgetting’ because, firstly, it requires the exposition to the drug-associated environment in the absence of this compound and, secondly, because drug-directed behaviours can be reinstated by exposure to certain stimuli [11].

The retrieval and subsequent reconsolidation of opiate withdrawal memories requires synaptic and structural plasticity in the hippocampus and the basolateral amygdala (BLA), among other brain nuclei [15,16,17]. For these neuroplastic processes to occur are needed immediate early genes (IEGs) expression and protein synthesis [18]. Extinction memories, as consequences of a new associative learning process, have also been suggested to require protein synthesis for their consolidation and recall [12]. 

The mammalian (or mechanistic) target of Rapamycin (mTOR) complex I (mTORC1) is a Thr/Ser protein kinase that participates in the formation of long-term memory (LTM) through the regulation of dendritic synaptic proteins synthesis [19,20]. In addition, mTORC1 has been postulated recently as a central key for the neuroadaptations induced by drugs of abuse that result in the detrimental behaviours characteristic of addiction [20]. Among the targets proposed for mTORC1 [20] are the activity-regulated cytoskeleton-associated (Arc) protein, which participates in the consolidation and retrieval of CPA-related memories in the amygdala and hippocampus [16,17], the subunit 1 (GluN1) of the ionotropic glutamate receptor N-methyl-D-aspartate (NMDA)-type, which are needed for the reconsolidation and extinction of morphine withdrawal memories [21,22], and the postsynaptic scaffolding protein Homer, which regulates mGlu1 and 5 signalling at the postsynaptic density (PSD) and has been related with addiction to several drugs of abuse, learning and memory, and behaviour [23,24,25]. Hence, in the present work, we used the CPA paradigm to study the activity of mTORC1 during the retrieval and extinction of aversive memories associated with morphine withdrawal and the expression of synaptic proteins regulated by this pathway in the BLA and the dentate gyrus (DG) of the hippocampus. 

## 2. Materials and Methods

### 2.1. Animals

Wistar male adult rats (220–240 g at the beginning of the experiment) were housed in methacrylate cages (length: 45 cm; width: 24 cm; height: 20 cm; 2–3 rats per cage) under a 12 h light/dark cycle (light: 8:00–20:00 h) in a room with controlled temperature (22 ± 2 °C). Food and water were available ad libitum. Animals were conditioned and tested during the light phase of the cycle. They were handled daily during the first week after arrival to minimise stress. All surgical and experimental procedures were performed in accordance with the European Communities Council Directive of 22 September 2010 (2010/63/UE) and were approved by the local Committees for animal research (Comité de Ética y Experimentación Animal; CEEA; RD 53/2013).

### 2.2. Drugs

Morphine base was obtained from Alcaliber Laboratories (Madrid, Spain). Morphine was administered as pellets of sustained release containing morphine base (75 mg), Avicel (55 mg), polyvinylpyrrolidone (20 mg), Aerosil (0.75 mg), and magnesium stearate (1.5 mg). Placebo pellets contained the same compounds, but morphine base was replaced with lactose. Naloxone hydrochloride was purchased from Sigma Chemical (St. Louis, MO, USA), dissolved in sterile saline (0.9% NaCl; ERN Laboratories, Barcelona, Spain) and administered subcutaneously (s. c.). The dose of naloxone was 15 μg/kg and was injected in volumes of 1 mL/kg of body weight. This dose was selected given that it has been reported to evoke aversive emotional symptoms of opioid withdrawal and, consequently, elicit significant place aversion in morphine dependent animals but not in controls, and reduced physical ones [11,14,26]. 

### 2.3. Behavioural Procedures

#### 2.3.1. Induction of Morphine Dependence

Morphine dependence was induced by subcutaneous (s.c.) implantation of 2 morphine pellets in the interscapular area of the animals under isoflurane anaesthesia. This method has been proven to induce dependence within the next 24 h following the surgical procedure [27,28] and to maintain stable the plasmatic levels of morphine for 15 days [29,30]. Rats were randomly divided into two groups: one of them had lactose pellets implanted and the other group was surgically intervened with morphine pellets.

#### 2.3.2. Conditioning Apparatus

Conditioning apparatus (Panlab, Barcelona, Spain) consisted of a box separated in two same-size chambers (40 × 13 × 45 cm) connected through a rectangular corridor (25 × 13 × 45 cm). Both chambers show different visual patterns on the walls (black dots or grey stripes), different colour and texture of the floor (black or grey, smooth or rough, respectively). The combinations chosen were: (A) black-dotted walls, smooth black floor; and (B) black-stripped walls, rough grey floor. Walls in the corridor were transparent, which minimized the time that the animals stay in it. The position of the animal during the test and the number of entries in every chamber were detected through transduction technology and the program PPCWIN (Panlab). Experimental protocol consisted of three phases: pre-conditioning, conditioning, and test. Since chronic morphine treatment reduces weight gain because of a lower caloric intake [27,28,31,32], animal weight was measured every day to check that morphine was properly released from the pellets. 

#### 2.3.3. Conditioning Place Aversion Protocol (CPA)

Briefly, CPA protocol consisted of three phases (Figure 1A). Firstly, during the pre-conditioning phase, rats were allowed to explore freely the conditioning apparatus to test and exclude those with natural preference to any chamber. Secondly, in the conditioning phase, morphine withdrawal was induced by administration of naloxone and animals were confined to one of the compartments, which allowed them to associate the negative symptoms of withdrawal with that environment. In the last stage, animals were again allowed to freely explore the apparatus and to test whether they retrieved the environmental memories associated with the abstinence syndrome and, therefore, avoided the withdrawal-paired chamber. 

##### Pre-Conditioning Phase

In this phase (day 0), animals were placed in the central corridor and were free to explore the apparatus for 30 min (pre-test). Animals that showed natural preference or aversion for one of the chambers (more than 60% of the time and less of the 40% of the time of the session, respectively) were discarded. One chamber was randomly chosen for the animal to associate it with withdrawal syndrome to morphine, and the other was where the animal was placed after saline administration (Figure 1A).

##### Conditioning Phase

In this phase, guillotine doors blocked access from both compartments to the central corridor. Three days after pellets implantation, animals received a s.c. injection of saline and were confined in their previously assigned chamber for 1 h. Three hours after the saline administration, rats received a dose of naloxone s.c. to provoke an emotional withdrawal syndrome and were placed in the withdrawal syndrome opposite compartment for 1 h. This process was repeated for 2 consecutive days for control and morphine-treated rats (Figure 1A). 

##### CPA Test

CPA test was performed the following day after the last conditioning session, similarly to the pre-conditioning phase: animals had 30 min to explore freely both chambers. Sixty min after the CPA test started, part of the morphine-dependent animals (Morphine-CPA; M-CPA) and part of the controls (Placebo-CPA; P-CPA) were sacrificed by decapitation or transcardiac perfusion. Resulting scores of the difference between the time that animals stayed in the compartment associated with morphine-withdrawal during the CPA test and that during the pre-conditioning test were obtained (Figure 1A). 

#### 2.3.4. Extinction of the CPA Protocol 

##### Extinction Training Phase

In this phase, guillotine doors blocked access from both chambers to the central corridor. After CPA testing, another group of morphine dependent rats (Morphine-Extinction Training; M-ET) and their controls (Placebo-Extinction Training; P-ET) followed the extinction conditioning protocol of Myers et al. (2012) [11] with several modifications. The next day after CPA test, rats were injected with saline and placed in the chamber previously assigned to saline for 30 min. After this period, rats were put back to their cages. Three hours after the first injection, rats were injected again with saline and placed in the opposite chamber, previously associated with withdrawal syndrome, for 30 min. This process was repeated for 3 days. After the extinction session of the third day, a set of animals were sacrificed through decapitation (Figure 1A). 

##### Extinction of the CPA Test

Extinction test was carried out similarly to the pre-conditioning and CPA tests. Control (Placebo-Extinction; P-EXT) and morphine-treated (Morphine-Extinction; M-EXT) rats were free to explore both compartments for 30 min. Sixty min after starting the test, animals were sacrificed through decapitation or transcardiac perfusion. Resulting scores of the difference between the time animals stayed in the chamber associated with morphine-withdrawal during the extinction test and the pre-conditioning test were calculated (Figure 1A).

### 2.4. Sample Processing

After decapitation, brain was rapidly extracted for quantification, through immunoblotting, of p-mTOR and mTOR, and for determination of mRNA through RT-qPCR of Arc, Homer1, and GluN1. Coronal sections of the brain (500 μm) were obtained through cryostat following the Paxinos and Watson (2007) atlas [33]. In the cryostat, brains were kept at −20 °C until the area of interest was in the plane of the section (BLA, bregma: −1.90 to −3.40 mm; DG, bregma: −3.30 to −3.40). Later, nuclei of interest were micropunched bilaterally with an instrument of 1 mm diameter and kept in tubes at −80 °C.

Another group of rats was anesthetized with a sublethal dose of pentobarbital (100 mg/kg intraperitoneal; i.p.) and a transcardiac perfusion with 250 mL of saline 0.9% followed by 500 mL of fixative solution (paraformaldehyde 4% in borate buffer 0.1 M pH 9.5) was performed. After extraction, brains were kept in the same fixative solution but including sucrose (30%) for 3 h.

### 2.5. Electrophoresis and Immunoblotting 

Three micropunches of BLA or DG were homogenized following the Beldjoud et al. (2016) [34] protocol. Cytoplasmatic fraction of the sample was isolated, and protein concentration was determined by using the bicinchoninic acid (BCA) method. Fifteen micrograms of total protein were loaded in a polyacrylamide gel (7.5%, BioRad Laboratories, Hercules, CA, USA), electrophoresis was performed, and proteins were transferred to a polyvinylidene difluoride (PVDF; Millipore, Bedford, MA, USA). Membranes were blocked with bovine serum albumin (1%) in TBS buffer with Tween 20 (0.15%) for 60 min at room temperature. Immunoblotting analysis was performed with the following monoclonal primary antibodies: rabbit anti p-mTOR (1:1000; #5536, Cell Signaling Technology, Danvers, MA, USA) and rabbit anti mTOR (1:1000; #2983, Cell Signaling Technology). After three washings with TBST (tris buffer saline tween, 0.15%), the membranes were incubated 1 h at room temperature with the peroxidase-labelled polyclonal secondary anti-rabbit antibody (#31430; 1:10,000, Thermo Fisher Scientific, Waltham, MA, USA). After washing, immunoreactivity was detected with an enhanced chemiluminescent immunoblotting detection system (ECLPlus, Thermo Fisher Scientific, Waltham, MA, USA) and visualized by a LAS 500 (GE Healthcare, Boston, MA, USA) Imager. Antibodies were stripped from the blots by incubation with stripping buffer (glycine 25 mM and SDS 1%, pH 2) for 1 h at 37 °C. The integrated optical density of the bands was normalized to the background values. Relative variations between bands of the experimental samples and the control samples were calculated in the same image. The ratio p-mTOR/mTOR was plotted and analysed. 

### 2.6. RNA Extraction and Quantitative Real-Time PCR (RT-qPCR)

One micropunch of both nuclei was homogenized with Quiazol (Qiagen, Valencia, CA, USA) and total RNA was extracted with the Qiagen RNeasy Lipid Tissue Mini Kit (Qiagen). For this purpose, manufacturer’s instructions were followed. RNA concentration was measured in a spectrophotometer NanoDrop (Thermo Fisher Scientific). cDNA synthesis was performed with the High Capacity cDNA Reverse Transcription (Applied Biosystems, Waltham, MA, USA). To avoid RNA degradation, RNAase inhibitors (Applied Biosystems) were used at a final concentration of 1.0 U/μL. qPCR primers were designed with Primer3 software (Whitehead Institute, Cambridge, MA, USA). Primers (Table 1; Integrated DNA Technologies, Leuven, Belgium) were used in qPCR with SybrGreen qPCR Master Mix (Applied Biosystems). qPCR experiments were carried out in the Fast Real-Time PCR System^®®^ apparatus (Applied Biosystems). Amplifications were carried out in triplicate and the relative expression of target genes was determined by the ΔΔCT method.

### 2.7. Immunofluorescence Assays

#### 2.7.1. pS6-GLS2 and pS6-GAD Labelling

In order to identify the brain regions of interest, Paxinos and Watson atlas (2007; [33]) was used. Sections were washed with PBS and an antigen retrieval technique was carried out. To accomplish this, sections were exposed to citrate buffer (citric acid 10 mM in 0.05% of Tween-20, pH 6.0, 90 °C; 2 times for 10 min each). Unspecific bindings were blocked with BSA solution (5% in 0.3% of Triton-X-100 in PBS for 90 min, RT). Two assays were performed. For pS6-GLS2 double-labelling, sections were incubated for 72 h at 4 °C with gentle agitation with rabbit polyclonal anti GLS2 (ab113509; 1:1000, Abcam, Cambridge, UK) and sheep polyclonal anti pS6 (ab65748; 1:250, Abcam) antibodies. For pS6-GAD double-labelling, chicken polyclonal anti GAD (NBP1-02161; 1:750, Novus Biologicals, Centennial, CO, USA) and sheep polyclonal anti pS6 (ab65748; 1:250, Abcam) were used. After this, brain tissue samples were incubated with fluorochrome-conjugated secondary antibodies for 4 h. For pS6-GLS2 staining, a polyclonal anti-rabbit antibody conjugated with Alexa Fluor 488 (A-11015; 1:1000, Invitrogen, Waltham, MA, USA) and a polyclonal anti-sheep antibody conjugated with Alexa Fluor 555 (A-31572; 1:1000, Invitrogen) were used. For pS6-GAD staining, a polyclonal anti-sheep antibody conjugated with Alexa Fluor 555 (A-31572; 1:1000, Invitrogen) and a polyclonal anti-chicken antibody conjugated with Alexa Fluor 647 (A-21449; 1:1000, Invitrogen) were used. Sections were also incubated with DAPI (1:25,000) for 1 min and mounted with ProLong Gold (Invitrogen) in gelatinized slides.

#### 2.7.2. Arc, GluN1, and Homer1 Triple Labelling

After the selection of DG- and BLA-containing tissue sections, washings with PBS and antigen retrieval technique proceedings were carried out as described previously. Unspecific bindings were blocked with BSA solution (3% in 0.3% of Triton-X-100 in PBS for 90 min, RT). Sections were incubated for 48 h at 4 °C with gentle agitation with guinea pig polyclonal anti Arc (156 004; 1:1500; Synaptic Systems, Goettingen, Germany), mouse monoclonal anti GluN1 antibody (32-0500; 1:100; Thermo Fischer Scientific) and rabbit monoclonal anti Homer1 (ab184955; 1:150; Abcam). Later, brain tissue samples were incubated with the following secondary antibodies: anti-mouse conjugated with Alexa Fluor 555 (A-31570, 1:1000, Invitrogen), anti-guinea pig conjugated with Alexa 488 (A-11073, 1:1000, Invitrogen), and anti-rabbit conjugated with Alexa 647 (A-31573; 1:1000, Invitrogen) for 4 h, as described in the previous section. Sections were also incubated with DAPI (1:25,000) for 1 min and mounted with ProLong Gold (Invitrogen) in gelatinized slides.

### 2.8. Confocal Colocalization Analysis of pS6-GLS2 and pS6-GAD Double Labelling

Images were captured by using a confocal microscope Leica TCS SP8 (Leica Microsystems, Barcelona, Spain) and LAS X (Leica Microsystems) processing software. Images from the nuclei were captured from low magnification to high magnification (10× to 63× with immersive oil for BLA, 10× to 40× for DG). Confocal images were obtained using 405 nm excitation for DAPI, 488 nm excitation for Alexa Fluor 488, 555 nm excitation for Alexa Fluor 555 and 647 nm excitation for Alexa Fluor 647. Emitted light was detected in the range of 405–490 nm for DAPI, 510–550 nm for Alexa Fluor 488, 555–640 nm for Alexa Fluor 555, and 647–755 nm for Alexa Fluor 647. Every channel was captured separately to avoid spectral cross-talking. The confocal microscope settings were stablished and maintained by local technicians for optimal resolution.

### 2.9. Quantitative Analysis of Arc-GluN1-Homer1 Triple Labelling

Images were captured by using a Leica epifluorescence microscope (Leica DM4 B) connected to a video camera (Leica DFC7000 T). Time exposure (2 s) and settings for both nuclei were constant through experimental groups, and images were captured by a blinded investigator at 20x magnification. Quantification of the images was performed by using FIJI software v. 2.1.0/1.53c (NIH ImageJ, Bethesda, MD, USA). Firstly, “.lif” documents exported from LAS X were opened as “hyperstack” through the Bio-format plugin; then, the region of interest corresponding to BLA and DG was selected manually in one of the channels and the same region was replicated automatically in the following captured channels. Afterwards, the mean grey value of these regions was measured automatically by this software. Three to six sections of each animal (*n* = 4–5 animals per group) were evaluated and a mean value for each animal was then calculated.

### 2.10. Statistical Analysis

Data were analysed using GraphPad Prism 9.0 (GraphPad Software; San Diego, CA, USA), and *p*-values < 0.05 were considered statistically significant. All descriptive data were presented as means ± standard error of the mean (SEM). Results of behavioural tests, immunoblotting and triple-labelling quantification were analysed using Student’s *t* test or two-way analysis of variance (ANOVA) followed by Bonferroni post hoc test. All our variables were continuous and paired. We confirmed the absence of outliers (ROUT Method; Q = 1%) and performed several normality and lognormality tests (Anderson–Darling, D’Agostino and Pearson, Shapiro–Wilk, and Kolmogorov–Smirnov tests), that were passed by all the variables. Thus, given the Gaussian distribution of our data, we computed the Pearson correlation to explore whether there was a linear association between some variables and its strength. 

## 3. Results

It is known that chronic opiate exposure provokes a lower weight gain due to a lesser caloric intake [27,28,31,32]. While no significant differences (t94 = 0.7854; *p* = 0.4342) were found in the weight gained by animals in the 5 days previous to pellets implantation (from day −4 to day 1) that were posteriorly treated with placebo (27.29 ± 1.13 g; *n* = 49) or morphine (28.87 ± 1.69 g; *n* = 47), in agreement with the aforementioned studies, Student’s *t* test uncovered that, from the day of pellets implantation (day 1) to the first day of conditioning (day 4), morphine-treated rats enhanced significantly (t94 = 8.2530; *p* < 0.0001) less their body weight (7.28 ± 1.49 g; *n* = 47) than the placebo group (22.59 ± 1.12 g; *n* = 49).

### 3.1. Extinction Training Suppressed the Aversive Behaviour Induced by Opiate Withdrawal Syndrome

After two consecutive days of conditioning, rats were tested for aversion to the naloxone-paired chamber and some of them were posteriorly sacrificed. Student’s *t* test revealed that the score of the morphine-treated rats that were sacrificed after the CPA test was significantly (t43 = 4.790, *p* < 0.0001) lower than the score of control animals sacrificed at the same time point (Figure 1B), thus indicating that the dose of naloxone administered for the conditioning elicited the aversive emotional state characteristic of morphine withdrawal in opiate-dependent rats but not in controls. 

Another group of rats followed a reconditioning procedure in order to extinguish the previously acquired CPA to the morphine withdrawal-associated chamber. Two-way ANOVA analysis of the scores showed a significant influence of the pharmacological treatment (F (1, 40) = 11.08; *p* = 0.0019) and the behavioural procedures (F (1, 40) = 16.43; *p* = 0.0002), but not of their interaction (F (1, 40) = 1.318; *p* = 0.2578). Bonferroni post hoc test confirmed that the CPA score of animals treated with morphine was significantly lower than the CPA score of the placebo group and, additionally, exhibited that the extinction score of opiate-dependent animals was significantly higher than their previous CPA score and statistically not different than the CPA and extinction scores of control animals. Hence, the reconditioning procedure was effective to suppress the CPA induced by morphine withdrawal (Figure 1B).

### 3.2. Morphine Withdrawal-Induced CPA Decreased mTOR Phosphorylation in the DG

Our experiments addressed the implication of mTORC1 on the retrieval of aversive memories of morphine withdrawal (CPA test) as well as on their extinction (extinction test). For that, first we measured phosphorylated (p)- and total-mTOR levels in the BLA and hippocampal DG and calculated mTOR phosphorylation ratio (p-mTOR/mTOR). It must be taken into consideration that this ratio is not a marker of mTOR nor p-mTOR levels, but an index of mTOR phosphorylation.

In the BLA, Student’s *t* test did not reveal significative differences (t10 = 1.635; *p* = 0.1331) in the ratio p-mTOR/mTOR of placebo and morphine-treated animals after the CPA test (Figure 2A). To compare the data of animals that showed CPA vs. animals that extinguished it, we used two-way ANOVA followed by a Bonferroni’s post hoc test. Two-way ANOVA failed to show main effects of the pharmacological treatment (F (1, 25) = 2.233; *p* = 0.1476), the behavioural procedures (F (1, 25) = 0.2675; *p* = 0.6095) or their interaction (F (1, 25) = 0.5220; *p* = 0.4767) on mTOR phosphorylation in the BLA (Figure 2A) after CPA extinction.

In contrast, in the hippocampal DG Student’s *t* test manifested a significant (t13 = 3.176, *p* = 0.0073) decrease in p-mTOR/mTOR ratio after CPA in morphine dependent animals regarding their controls (Figure 3A). Nonetheless, two-way ANOVA of p-mTOR/mTOR failed to exhibit main effects of the pharmacological (F (1, 27) = 0.9861; *p* = 0.3295) and the behavioural (F (1, 27) = 2.227; *p* = 0.1472) factors nor their interaction (F (1, 27) = 1.283; *p* = 0.2673) after extinguishing naloxone-induced CPA (Figure 3A). 

In our experiments we also evaluated the involvement of mTOR in the process of reconditioning to a neutral stimulus (extinction training) after having developed an aversion for the morphine withdrawal-associated environment. For that, a set of morphine-dependent and control rats were sacrificed after the second reconditioning session on the third day of extinction training (day 9). Nonetheless, Student’s *t* test did not show significant differences in the ratio p-mTOR/mTOR in the BLA (t14 = 0.6947, *p* = 0.4986; Figure 2B) nor the DG (t15 = 1.056, *p* = 0.3078; Figure 3B) of morphine-treated animals compared with that of the placebo group.

### 3.3. Characterization of mTORC1-Expressing Neurons

mTOR, by binding to distinct proteins, can form two different complexes, of which mTORC1 is broadly known to participate both in the adaptive alterations induced by drugs of abuse in the brain and in various learning and memory processes (19, 20). Hence, we next studied, by means of immunofluorescence, the neuronal populations in which mTORC1 was activated in BLA and DG after morphine withdrawal-induced CPA and after CPA extinction. For that, we colocalized phosphorylated S6 (pS6; as a marker of mTORC1 pathway activity) with glutaminase 2 (GLS2; involved in glutamate synthesis) and glutamate decarboxylase (GAD; an enzyme that participates in GABA synthesis). Our study showed that both glutamatergic and GABAergic neurons in the BLA expressed pS6 (Figure 1C). Similarly, pS6 colocalized with GLS2 and GAD in the DG (Figure 2C). 

### 3.4. The Extinction of Morphine Withdrawal-Induced CPA Increased Arc and GluN1 Expression in the DG and the BLA

Then, we investigated the differential expression of some mTORC1 targets (Arc, GluN1, and Homer1) in the BLA and DG after the expression of morphine withdrawal-induced CPA and after its extinction.

In the BLA, Student’s *t* test exhibited that morphine withdrawal-induced CPA significatively (t8 = 2.580, *p* = 0.0326; Figure 4A) increased Arc-immunoreactivity (IR; measured as the mean grey value) but did not alter GluN1-IR (t8 = 0.2960, *p* = 0.7748; Figure 4D) and Homer1-IR (t8 = 0.3139, *p* = 0.7616; Figure 4G). On the other hand, after the extinction test two-way ANOVA manifested main effects of the pharmacological treatment for Arc-IR (F (1, 16) = 10.02; *p* = 0.0060) but not for GluN1-IR (F (1, 16) = 0.4134; *p* = 0.5294) nor Homer1 (F (1, 16) = 0.4142; *p* = 0.5290), and the behavioural procedures for Arc-IR (F (1, 16) = 8.160; *p* = 0.0114), GluN1-IR (F (1, 16) = 20.39; *p* = 0.0004), and Homer1-IR (F (1, 16) = 4.778; *p* = 0.0440). Nonetheless, two-way ANOVA failed to detect significant effects of the interaction between the pharmacological and behavioural factors for Arc-IR (F (1, 16) = 0.3269; *p* = 0.5754), GluN1-IR (F (1, 16) = 1.011; *p* = 0.3297), and Homer1-IR (F (1, 16) = 0.9616; *p* = 0.3414). Bonferroni’s post hoc test confirmed that the reconsolidation of morphine withdrawal memories significantly increased Arc-IR in morphine-treated rats regarding control animals (Figure 4A). Additionally, the post hoc test manifested that after the extinction test there was a significant increase in GluN1-IR in control and morphine-dependent animals regarding the CPA test, respectively (Figure 4C). 

In the DG, Student’s *t* test showed that morphine withdrawal-induced CPA significantly increased Arc-IR (t8 = 3.728, *p* = 0.0058; Figure 5A) and GluN1-IR (t8 = 4.142, *p* = 0.0032; Figure 5C), and that these proteins were mainly located in granule cells (Figure 5G). Nevertheless, we did not find significant changes in Homer1-IR (t8 = 0.3694, *p* = 0.7214). On the other hand, after the extinction test, two-way ANOVA exposed main effects of the pharmacological treatment for Arc-IR (F (1, 15) = 7.062; *p* = 0.0179), but not for GluN1-IR (F (1, 15) = 0.4090; *p* = 0.5321) nor Homer1 (F (1, 15) = 0.1929; *p* = 0.6668). Two-way ANOVA uncovered a significant effect of the behavioural procedures as well for Arc-IR (F (1, 15) = 9.924; *p* = 0.0066) and GluN1-IR (F (1, 15) = 17.12; *p* = 0.0009), but not for Homer1-IR (F (1, 15) = 0.7745; *p* = 0.3927). Finally, ANOVA detected a main effect of the interaction of the pharmacological treatment X behavioural procedures for GluN1-IR (F (1, 15) = 6.687; *p* = 0.0207), but not for Arc-IR (F (1, 15) = 1.018; *p* = 0.3289) nor Homer1-IR (F (1, 15) = 0.7745; *p* = 0.3927). Bonferroni’s test also revealed that morphine-treated rats exhibited significantly higher Arc-IR than placebo-treated rats after the CPA test (Figure 5A). Moreover, post hoc tests showed that Arc-IR and GluN1-IR in the DG of control rats after the extinction test were statistically enhanced when compared to the same group of animals after the CPA test (Figure 5A,C).

Next, we evaluated whether there was any correlation between the expression of any of these proteins and the extinction of morphine withdrawal-induced CPA quantified as the difference in time that rats spent in the naloxone-paired chamber in the extinction test minus that in the CPA test. We uncovered an almost perfect negative correlation between GluN1-IR and this difference in time in placebo-treated animals in the BLA (r = −0.9391, *p* = 0.0171; Figure 4G) and in the DG (r = −0.9980, *p* = 0.0020; Figure 5H). On the other hand, in morphine-dependent rats Homer1-IR almost perfectly correlated negatively with the difference in time that animals spent in the naloxone-paired chamber in the extinction and in the CPA tests in the BLA (r = −0.9082, *p* = 0.0329; Figure 4H) and in the DG (r = −0.9672, *p* = 0.0071; Figure 5I). In addition, Arc-IR very highly correlated negatively with this difference in time exclusively in the DG of opiate-treated animals (r = −0.8997, *p* = 0.0375; Figure 5I). Lastly, as can be seen in Table 2, when all the animals (placebo- and morphine-treated) were considered, we detected from high to very high negative correlations between the subtraction of times spent in the withdrawal-associated compartment in the extinction and CPA tests and, on the one hand, GluN1 in the BLA (r = −0.7176, *p* = 0.0195) and in the DG (r = −0.7384, *p* = 0.0231) and, on the other hand, Homer1-IR in the BLA (r = −0.6923, *p* = 0.0265; Figure 4J) and the DG (r = −0.7204, *p* = 0.0286).

Finally, we studied the correlation between the expression of each of these proteins (Table 3). We unmasked from very high to almost perfect positive correlations between Arc-IR and GluN1-IR in control animals after the CPA test in the BLA (r = 0.8853, *p* = 0.0458) and in the DG (r = 0.9487, *p* = 0.0139) and in morphine-dependent animals after the extinction test in the BLA (r = 0.9581, *p* = 0.0102). Arc-IR also almost perfectly correlated positively with Homer1-IR in the BLA (r = 0.9312, *p* = 0.0214) and the DG (r = 0.9248, *p* = 0.0245) of morphine-treated animals after the extinction test. Lastly, GluN1-IR and Homer1-IR correlated positively almost perfectly in the DG of opiate-dependent animals after the CPA test (r = 0.9358, *p* = 0.0193). When all the experimental conditions were considered, we found from high to very high positive correlations between Arc-IR and GluN1-IR in the BLA (r = 0.6772, *p* = 0.0010) and the DG (r = 0.7225, *p* = 0.0050) and between Arc-IR and Homer1-IR in the BLA (r = 0.6803, *p* = 0.0010) and the DG (r = 0.6187, *p* = 0.0470). Ultimately, we saw a moderate positive correlation between GluN1-IR and Homer1-IR in the DG (r = 0.5543, *p* = 0.0160). 

To study whether the alterations in the expression of mTORC1 targets in the BLA and DG were due to modified translation of these proteins or to transcriptional mechanisms, we determined mRNA levels of *Arc*, *Grin1,* and *Homer1*. Student’s *t* test did not show significant changes in *Arc* (t8 = 0.01962, *p* = 0.9848; Figure 4B), *Grin1* (t10 = 1.428, *p* = 0.1839; Figure 4D), nor *Homer1* (t9 = 0.2819, *p* = 0.7844; Figure 4F) mRNA levels in the BLA of morphine dependent animals showing withdrawal-induced CPA in comparison with controls. Two-way ANOVA also failed to reveal main effects of the pharmacological (*Arc*: F (1, 16) = 2.162, *p* = 0.1609; *Grin1*: F (1, 19) = 0.9370, *p* = 0.3452; *Homer1*: F (1, 17) = 0.006824, *p* = 0.9351) and behavioural (*Arc*: F (1, 16) = 0.08410, P = 0.7755; *Grin1*: F (1, 19) = 2.020, *p* = 0.1714; *Homer1*: F (1, 17) = 3.147, *p* = 0.0940) nor their interaction (*Arc*: (F (1, 16) = 2.245, *p* = 0.1535); *Grin1*: (F (1, 19) = 1.397; *p* = 0.2518); *Homer1*: (F (1, 17) = 0.1748, *p* = 0.6811) on mRNA levels after CPA extinction (Figure 4B,D,F). In the DG, although a clear tendency to augment in Arc mRNA levels of morphine treated rats after the retrieval of morphine withdrawal-paired memories can be seen, Student’s *t* test failed to exhibit significant (*Arc*: t9 = 1.757, *p* = 0.1127; *Grin1*: t9 = 0.6362, *p* = 0.5405; *Homer1*: t9 = 1.394, *p* = 0.1966) differences in any of the mRNAs studied in opiate-treated animals after the CPA test regarding the controls (Figure 5B,D,F). Following the extinction test, two-way ANOVA did not manifest significant effects of the pharmacological (*Arc*: F (1, 16) = 1.543, *p* = 0.2321; *Grin1*: F (1, 18) = 0.08037, *p* = 0.7800; *Homer1*: F (1, 18) = 0.9161, P = 0.3512) and behavioural (*Arc*: F (1, 16) = 3.586, *p* = 0.0765; *Grin1*: F (1, 18) = 1.901, *p* = 0.1848; *Homer1*: F (1, 18) = 2.229, *p* = 0.1528) treatments nor their interaction (*Arc:* (F (1, 16) = 3.222, *p* = 0.0915); *Grin1* (F (1, 18) = 1.507, *p* = 0.2354); *Homer1* (F (1, 18) = 1.505, *p* = 0.2358) on mRNA levels in the DG (Figure 5B,D,F). 

Despite the lack of changes in Arc mRNA relative levels, there was an almost perfect negative correlation between Arc mRNA and the CPA score in the DG (r = −0.9488, *p* = 0.0138; Figure 5G). In addition, considering all the animals (placebo- and morphine-treated) after the extinction test, we did observe a high positive correlation between Homer1 mRNA levels in the BLA and the difference in time that these animals spent in the naloxone-associated compartment during extinction and CPA tests (r = 0.6936, *p* = 0.0261; Figure 4I). 

We then analysed whether there were any correlations between the mRNA levels of each of these proteins (Table 4). In placebo-treated animals after the CPA test we detected from very high to almost perfect positive correlations between Arc and Grin1 relative mRNA levels in the BLA (r = 0.9536, *p* = 0.0119) and the DG (r = 0.9902, *p* = 0.0001), between Arc and Homer1 mRNAs in the BLA (r = 0.9796, *p* = 0.0204) and the DG (r = 0.9117, *p* = 0.0114) and between Grin1 and Homer1 mRNAs in the DG (r = 0.8715, *p* = 0.0237). After the extinction test, we found almost perfect positive correlations between Arc and Homer1 mRNAs (r = 0.9902, *p* = 0.0012) in the DG of control animals and between Grin1 and Homer1 mRNAs (r = 0.9004, *p* = 0.0144) in the DG of morphine-treated rats. Finally, when all the experimental conditions were analysed together, we found a moderate positive correlation between Arc and Homer1 mRNAs (r = 0.5345, *p* = 0.0104) and between Grin1 and Homer1 mRNAs (r = 0.6622, *p* = 0.0008) in the DG.

## 4. Discussion

In agreement with previous literature [14,16,17], the morphine-dependent animals used in this study developed CPA to the withdrawal-associated environment. In parallel, we observed a diminution in mTOR phosphorylation in the DG accompanied by an increase in the expression of Arc and GluN1. Accordingly, Arc mRNA levels in this area almost perfectly correlated negatively with the CPA score. In contrast, mTOR phosphorylation did not change in the BLA of morphine-treated rats following CPA. Nevertheless, the retrieval of morphine withdrawal memories did augment the expression of Arc in this region, pointing out that the mechanisms underpinning the recall of drug-associated aversive memories differ depending upon the brain nuclei. On the other hand, although mTOR has been reported to participate in memory consolidation [35], we did not find alterations in mTOR phosphorylation in the BLA nor the DG after the training in order to extinguish the previously acquired CPA to the withdrawal-paired compartment. Following the CPA extinction test, basal levels of mTOR phosphorylation were accompanied by the increased expression of Arc and GluN1 in the DG. In the BLA, GluN1 levels were also enhanced concomitantly with no alterations in mTOR activity, suggesting therefore that this kinase did not participate in the recall of opiate withdrawal extinction memories. Thus, mTOR-independent mechanisms would contribute to modulate the expression of Arc, GluN1, and Homer1 in these nuclei, which are vital for the extinction of withdrawal memories as shown by their almost perfect correlation with the extinction rate in the BLA and/or DG. 

It is well known that addicts associate the effects of drugs of abuse with environmental cues and, for this to occur, the same brain systems that mediate physiological learning and memory processes become abnormally activated [4,5]. The retrieval and later reconsolidation of withdrawal memories have been reported to activate the BLA and the hippocampus, altering the activity of some transcription factors, such as Arc and the phosphorylated cAMP response element binding protein (pCREB) [16,17,36,37]. The mTOR pathway intervenes in the regulation of these factors for the formation of LTM as well as for the neuroadaptations induced by drugs of abuse [19,20]. In turn, mTOR activity can be modulated by several neurotrophins, such as the brain-derived neurotrophic factor (BDNF) [38], which is known to be implicated in mTORC1-dependent expression of Arc, GluN1, and Homer1 in dendrites [20]. 

Our previous work unmasked that BDNF levels in the DG of rats after the retrieval of morphine withdrawal aversive memories diminish slightly [17], which agrees with the reduced phosphorylation of mTOR in the DG observed in this study following naloxone-induced CPA. This decreased mTOR phosphorylation was due to the recall of aversive memories associated with morphine withdrawal and not to opiate exposure, given that, when other groups of morphine-treated rats were tested for CPA extinction, mTOR phosphorylation in their DG was similar to that of the controls. Although mTOR activity has been related with the late long-term potentiation (L-LTP) that underlies LTM [38], some investigations have described controversial findings after the inhibition of the mTORC1 pathway, such as augmented learning and synaptic plasticity in the hippocampus of mice with Angelman syndrome [39], increased neurite outgrowth [40], or improved cognitive and affective deficits [41]. Moreover, the recall of cocaine-induced conditioned place preference (CPP) has been associated with decreased p-mTOR in several areas of the limbic system [42] and, in line with this data, we observed reduced p-mTOR/mTOR in the DG and BLA of mice after the reinstatement of cocaine CPP [43]. 

Arc is an indicator of genomically activated neurons as a result of memory retrieval and has been described to be critical for fear memory reconsolidation [44]. We detected enhanced Arc expression in the DG, suggesting therefore that the activity of the mTORC1 pathway in this region during the CPA recall and posterior reconsolidation correlates negatively with protein synthesis. In agreement with the present study, our previous investigations uncovered augmented Arc protein and mRNA levels in the DG concomitantly with a negative correlation between Arc protein levels and the aversion score in the CPA test [17]. Concordantly, present data showed that the CPA score strongly correlated with Arc mRNA levels in the DG of morphine dependent animals, thus corroborating the vital role of this protein in morphine-withdrawal memory retrieval and reconsolidation in this area, as it had been suggested before [15,45].

The increased GluN1 expression in parallel with decreased mTOR activity in the DG after the CPA test might be as well a result of a negative relationship between the mTORC1 pathway activity and protein synthesis. Previously, Niere et al. [46] observed diminished mTORC1 activity concomitantly with increased GluN1 in dendritic membranes and postulated that low mTORC1 activity could function as a signal to augment dendritic membrane excitability. Given that NMDA receptors are vital for the reconsolidation of morphine withdrawal memories [21] and that GluN1 is an essential subunit for the functionality of these receptors [47], our data would agree with that postulate. 

In contrast to the DG, we found unaltered mTOR phosphorylation in the BLA after the retrieval of withdrawal memories that, on the other hand, concurs with unpublished data from our laboratory revealing no changes in BDNF expression in this area after morphine withdrawal-induced CPA. However, in agreement with our previous observations of increased Arc protein in the BLA after the CPA test in morphine dependent rats [16], and confirming the key role of this area in the negative motivational component of morphine withdrawal, in the present study, Arc-IR in the BLA of the opiate-treated animals increased after the CPA test, pointing out that the enhancement in Arc levels occurs independently on mTOR signalling, as has previously been reported [48]. 

mTORC1 participates in protein synthesis through the phosphorylation of P70 S6 kinase, which, in turn, phosphorylates and so activates the ribosomal protein S6, which has been used by us and others as a marker of mTORC1 pathway activity [49]. We have found that pS6 is expressed in all the glutamatergic and GABAergic neurons of the BLA and DG after the expression of opiate withdrawal-induced CPA and after its extinction, thus not allowing us to discriminate between the neuronal populations where the mTOR pathway might be acting. Conflicting findings have been reported about the significance of S6 phosphorylation on protein translation. While early investigations revealed a correlation between S6 phosphorylation and translation [50], later it was published that enhanced pS6 is not enough to initiate this process [51] or even a negative role of pS6 on global protein synthesis [52]. Present data do not support either a role for mTOR in the modulation of transcription for the recall and reconsolidation of morphine withdrawal contextual memories, as opposed to what has been previously suggested for the retrieval of other kinds of memories [38]. 

Several treatments for addiction are based on the extinction of drug memories that are known to trigger relapse in drug use [4,5]. For fear memories, the extinction training during the reconsolidation period is known to improve extinction learning and prevent fear recovery [53]. Albeit the mTOR pathway has been considered essential for the formation of several types of memories in the hippocampus and amygdala [20], the activity of mTOR did not change in any of the brain areas examined after the acquisition of new associative contextual memories with neutral stimuli. The BLA is involved in the codification of conditioned stimuli’s emotional value, therefore playing a critical role in goal directed behaviours [16,37,54]. Hence, it could be postulated that the BLA might not be recruited for the formation of this emotionally neutral extinction memory, which would explain the lack of changes in mTOR activity observed in this area. Intriguingly, mTOR activity in the DG did not change either for the codification of the withdrawal extinction memories, thus suggesting that this kinase might not be crucial in this process in any of these nuclei. 

Consistent with previous findings [11,14], after three days of extinction training morphine-dependent rats did not manifest CPA to the naloxone-paired compartment in the extinction test. The recall of fear and other extinction memories has been reported to alter mTOR activity in the hippocampus [55,56,57]. Nevertheless, mTOR phosphorylation was not altered in the DG of morphine-dependent animals after the extinction test. Hence, the extinction of contextual fear and opiate withdrawal memories might not fully recruit the same signalling pathways and/or brain areas. 

We did not observe alterations of mTOR activity in the BLA of morphine-dependent animals after the extinction test. Whereas it seems logical that the BLA would not be engaged in the formation of extinction memories given their lack of emotional value, this region did become activated after the recall of opiate withdrawal extinction memories, as demonstrated by the increased GluN1 expression in morphine-dependent and control animals.

In spite of the unaltered mTOR activity, the expressions of Arc and GluN1 were augmented in both the BLA and DG of placebo and morphine-treated animals, thus indicating that the retrieval of extinction memories also triggers what seem to be mTOR-independent structural and synaptic plastic changes in these areas. Importantly, our correlation analyses strongly suggest that, while the extinction rate would be determined by GluN1 expression in both the DG and BLA of control animals, Homer1 in both nuclei and Arc in the DG could be both indicators for the extinction rate of morphine withdrawal memories. It should be noted that our postulate does not oppose the findings indicating that NMDA receptors are required for the extinction of morphine withdrawal memories [22], as we also detected increased GluN1-IR in opiate-dependent animals after the extinction test in both the DG and BLA. 

It might be surprising that the correlations between the extinction rate of the animals and their Arc-, GluN1-, and Homer1-IR in both the BLA and the DG were negative. These proteins are known to be necessary for the synaptic processes that occur for the retrieval and reconsolidation of memories [18,25,48]. However, they are rapidly degraded [48,58]. Thus, their syntheses would be required to maintain their levels throughout these processes in order for them to be effective. Concordantly, we found almost identical high correlations between the extinction rate and Homer1 protein levels and between the extinction rate and Homer1 mRNA levels, but with an opposite sign, thus pointing out that Homer1 transcription increased in parallel with extinction, and concurred with other findings of reduced Homer1 mRNA in circumstances of synaptic and memory degradation [59]. Moreover, for LTP consolidation ribosomal loading on Arc RNA is known to be necessary to ensure sustained Arc translation [18].

Homer1 has three different transcripts generated through alternative splicing, of which Homer1a is an activity-expressed truncated form that acts as a dominant negative regulator of the constitutively expressed long variants of Homers, thus disrupting the interactions between the lasts and their effector proteins at the PSD and allowing spine remodelling and synaptic plasticity mechanisms [25]. Findings of different studies point out that Homer1a and Arc, which are expressed connectively after neuronal activation during associative learning in the hippocampus, need to function co-ordinately for the formation of associative memories [60,61,62]. The almost perfect correlation between Arc and Homer1 protein expression in the DG, but also in the BLA of morphine-dependent animals after the extinction test, could imply that this association might be indispensable not only in the hippocampus, but also in other brain areas for the extinction of opiate withdrawal memories. Homer1 and Arc mRNA levels also correlated very highly in the DG and BLA of control animals after CPA and/or extinction tests, thus strengthening this hypothesis for the recall of associative memories. 

While long Homers act as intermediators between mGluRs located at extra- and peri-synaptic areas and NMDA receptors at the PSD, allowing the modulation of NMDA-evoked currents by mGluRs, Homer1a disrupts these associations for scaffold remodelling, probably leading to a change in mGluR-NMDA receptor function that would trigger neuroplastic processes [25]. Our correlations data suggest that Homer1, or probably the shorter variant Homer1a, would participate somehow in the regulation of mGluR-NMDA receptor activity in the DG during the retrieval of memories. As opposed to the association of Homer1 and Arc activity, protein or mRNA levels of Homer1 and GluN1 did not correlate in the BLA, which may indicate that the role of Homer1 as a mediator in the regulation of NMDA receptors currents by mGluRs is restricted to the hippocampus for the retrieval and/or extinction of withdrawal memories.

Arc transcription can be regulated, among others, by NMDA receptors, second messengers, protein kinases, and transcription factors [18,48,63]. Additionally, Arc can be regulated post-transcriptionally by some of the same stimuli that modulate its transcription, meaning that Arc protein expression may increase without an enhancement in Arc mRNA levels [63]. The strong positive correlations between Arc mRNA and protein levels with those of GluN1 in both the DG and the BLA of control animals after the CPA test might indicate that the activation of NMDA receptors would stimulate Arc transcription and translation in these conditions. Nonetheless, it seems that other signalling molecules would join in for the enhancement of Arc expression in these regions during the retrieval of morphine withdrawal memories and in control animals after the extinction test, given the lack of correlation with GluN1 or Grin1. The same occurred in the DG after the extinction of withdrawal memories. However, GluN1 and Arc protein levels correlated highly in the BLA after the extinction test in morphine-treated animals, thus highlighting the role of NMDA receptors in this region as modulators of Arc expression for the extinction of drug-associated aversive memories. This particularity in Arc regulation in the BLA could be due to the specific function of this area in the processing of emotional memories. 

## 5. Conclusions

The present study provides evidence of differences in the function of the mTOR signalling pathways in the modulation of the expression and activity of synaptic molecules during the retrieval and extinction of aversive memories of opiate withdrawal in amygdalar and hippocampal regions. These results confirm Arc expression in the DG as an index of the aversion for the morphine withdrawal-associated memories and point to Homer1 in the DG, but also in the BLA, as an indicator of the level of extinction of these kinds of memories. Finally, our data support coordinated activity of some of these neuroplastic proteins for the extinction of morphine withdrawal memories in a regional-dependent manner. Understanding the plastic mechanisms in which these molecules take part will allow the development of therapeutic strategies able to lessen the rate of relapses induced by drug withdrawal-associated aversive memories.

## Figures and Tables

**Figure 1 biomedicines-10-00588-f001:**
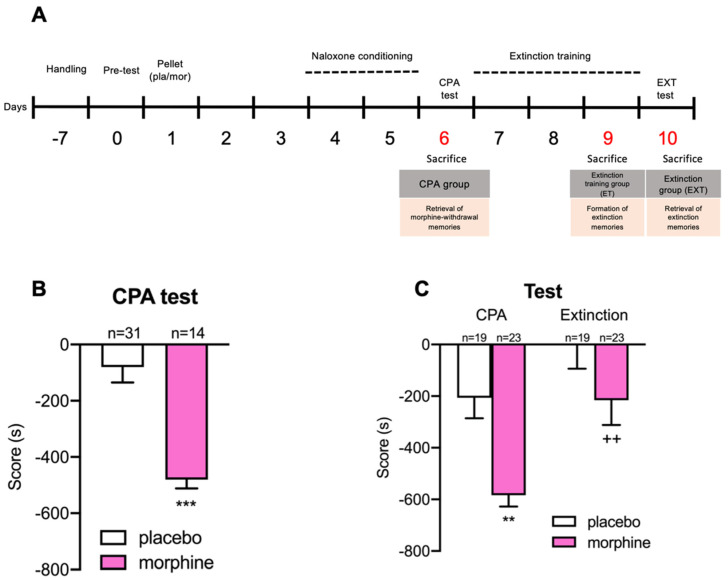
(**A**) Timeline of the behavioural procedures. After 7 days of habituation and handling, on day 0 animals were placed in the central corridor and allowed to explore the apparatus freely for 30 min (pre-test). On day 1, rats were implanted s.c. with 2 morphine or placebo pellets and were let to recover for 3 days. On day 4, for each rat, one chamber was randomly chosen to be paired with naloxone and the other chamber with saline (conditioning sessions). CPA test was conducted on day 6, exactly as in the preconditioning phase. After the test, for 3 days, rats were injected with saline and confined in both chambers. On day 10, rats were tested as in CPA test (extinction test). (**B**) CPA score. *** *p* < 0.001 vs. P-CPA. (**C**) CPA and extinction scores. ** *p* < 0.01, *** *p* < 0.001 vs. P-CPA; ++ *p* < 0.01 vs. M-CPA. Each bar corresponds to mean ± SEM of the mean.

**Figure 2 biomedicines-10-00588-f002:**
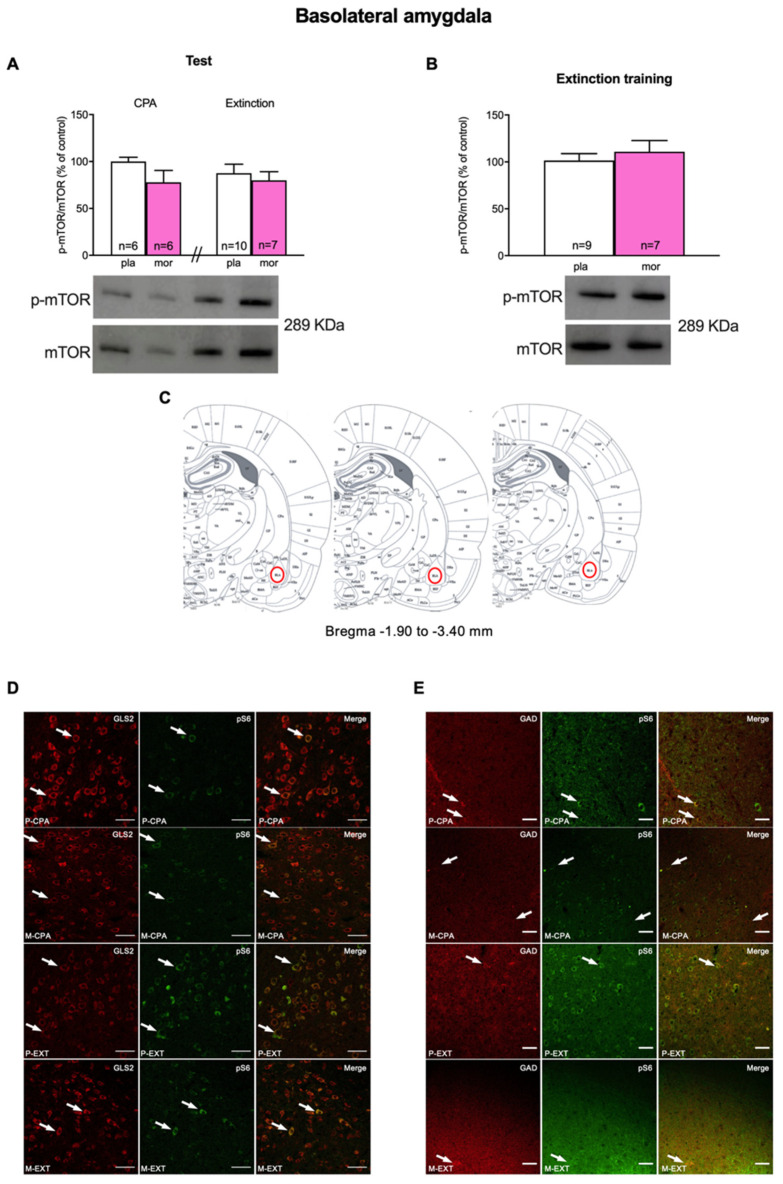
Basolateral amygdala: (**A**) p-mTOR/mTOR Western blot analysis after CPA and extinction tests. (**B**) pmTOR/mTOR analysis after extinction training procedures. (**C**) Anatomical localization of basolateral amygdala in the rat brain. (**D**) Colocalization of glutamatergic neurons (GLS2, red) and phosphorylated target of mTOR p-S6. (**E**) Colocalization of GABAergic neurons (GAD, red) and phosphorylated target of mTOR p-S6 (green). Each bar corresponds to mean ± SEM.

**Figure 3 biomedicines-10-00588-f003:**
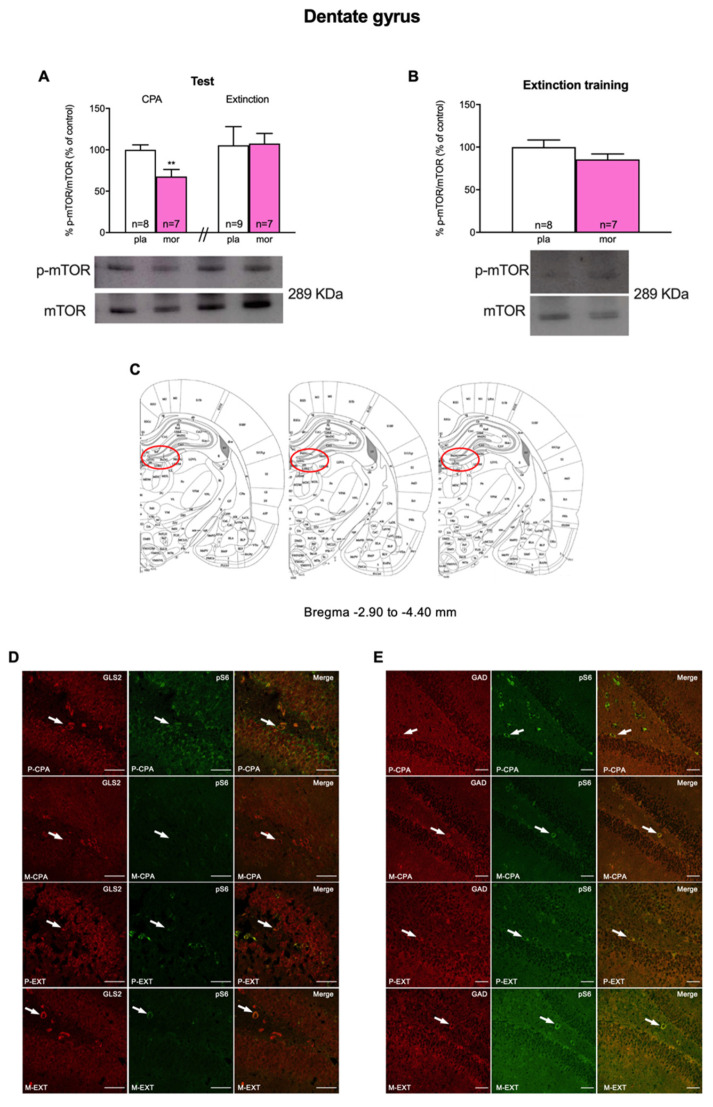
Dentate gyrus: (**A**) pmTOR/mTOR Western blot analysis after CPA and extinction tests. (**B**) pmTOR/mTOR analysis during extinction training procedures. (**C**) Anatomical localization of dentate gyrus in the rat brain. (**D**) Colocalization of glutamatergic neurons (GLS2, red) and phosphorylated target of mTOR p-S6 (green). (**E**) Colocalization of GABAergic neurons (GAD, red) and phosphorylated target of mTOR p-S6 (green). ** *p* < 0.01 vs. P-CPA. Each bar corresponds to mean ± SEM.

**Figure 4 biomedicines-10-00588-f004:**
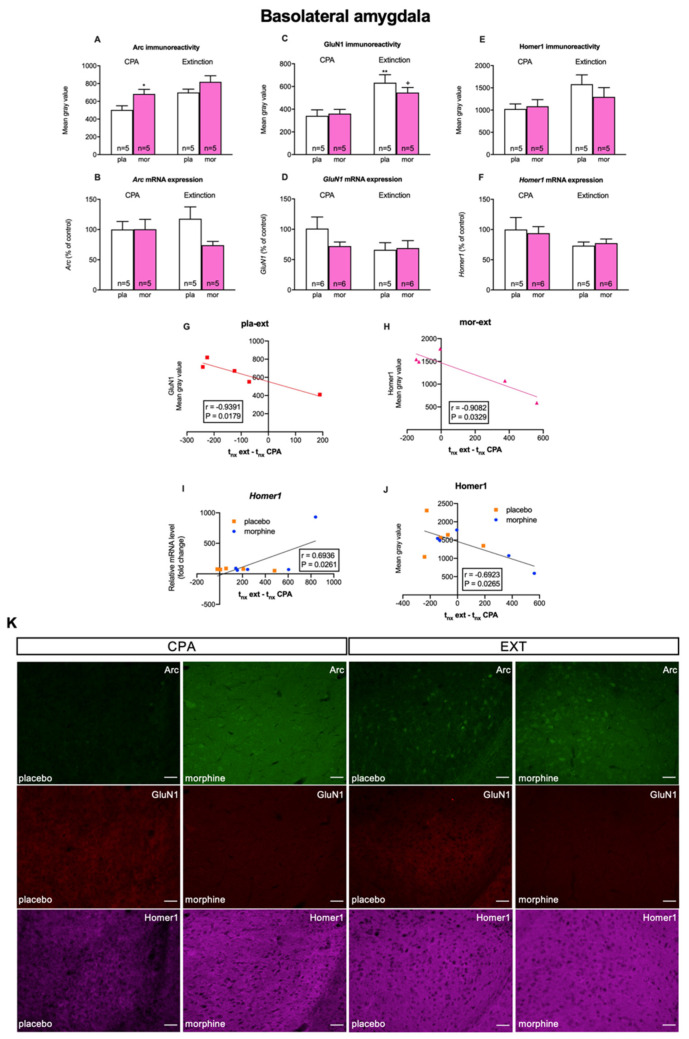
Basolateral amygdala: (**A**,**B**) Analysis of Arc immunoreactivity and mRNA expression during CPA and extinction test. (**C**,**D**). Analysis of GluN1 immunoreactivity and mRNA expression during CPA and extinction test. (**E**,**F**). Analysis of Homer1 immunoreactivity and mRNA expression during CPA and extinction test. (**G**) Correlation between GluN1 immunoreactivity and the difference between the time spent in the naloxone chamber during extinction test and CPA test in placebo rats that underwent extinction protocol. (**H**) Correlation between Homer1 immunoreactivity and the difference between the time spent in the naloxone chamber during extinction test and CPA test in morphine-treated rats that underwent the extinction protocol. (**I**) Correlation between relative Homer1 mRNA expression and the difference between the time spent in the naloxone chamber during extinction test and CPA test in placebo and morphine-treated rats that underwent extinction protocol. (**J**) Correlation between Homer1 immunoreactivity and the difference between the time spent in the naloxone chamber during extinction test and CPA test in placebo and morphine-treated rats that underwent extinction protocol. (**K**) Representative epifluorescence images showing triple immunostaining of Arc (green), GluN1 (red), and Homer1 (pink) during CPA and extinction test. * *p* < 0.05, ** *p* < 0.01 vs. P-CPA; + *p* < 0.05 vs. M-CPA. Each bar corresponds to mean ± SEM.

**Figure 5 biomedicines-10-00588-f005:**
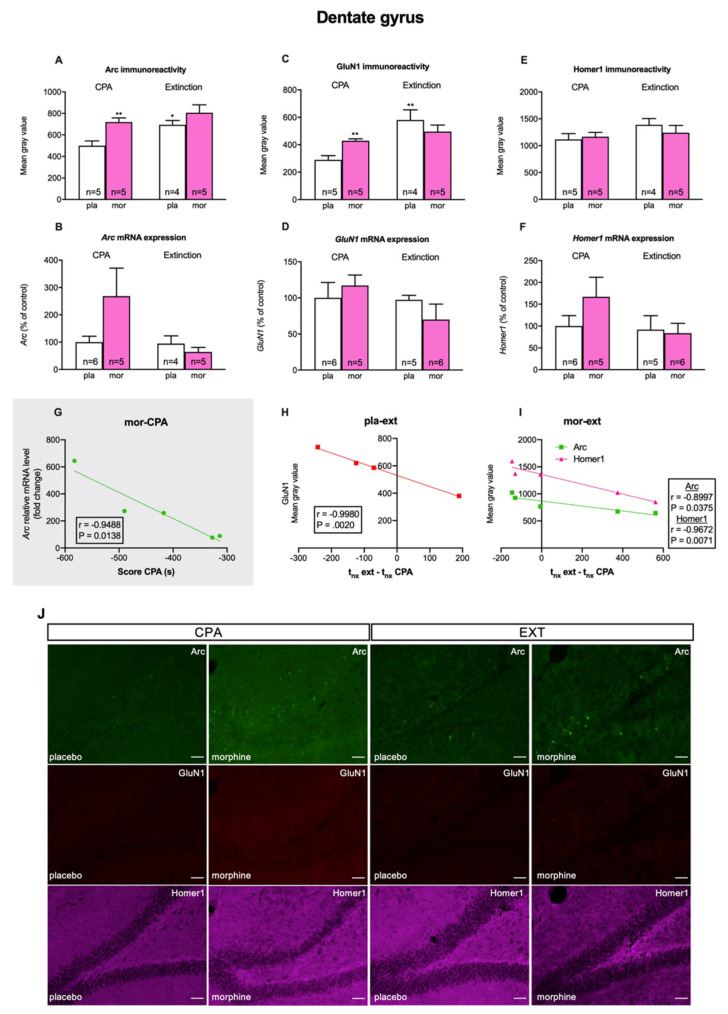
Dentate gyrus: (**A**,**B**) Analysis of Arc immunoreactivity and mRNA expression during CPA and extinction test. (**C**,**D**). Analysis of GluN1 immunoreactivity and mRNA expression during CPA and extinction test. (**E**,**F**). Analysis of Homer1 immunoreactivity and mRNA expression during CPA and extinction test. (**G**) Correlation between relative mRNA levels of Arc and the CPA score in morphine dependent rats. (**H**) Correlation between GluN1 immunoreactivity and the difference between the time spent in the naloxone chamber during extinction test and CPA test in placebo rats that underwent extinction protocol. (**I**) Correlation between Homer1 immunoreactivity and the difference between the time spent in the naloxone chamber during extinction test and CPA test in morphine-treated rats that underwent extinction protocol and between Arc immunoreactivity and same time parameter. (**J**) Representative epifluorescence images showing triple immunostaining of Arc (green), GluN1 (red) and Homer1 (pink) during CPA and extinction test. * *p* < 0.05, ** *p* < 0.01 vs. P-CPA. Each bar corresponds to mean ± SEM.

**Table 1 biomedicines-10-00588-t001:** Primers used in the qPCR experiments.

Gene	Forward	Reverse
β-Actin	CCCTAGACTTCGAGCAAGAGATG	CCACAGGATTCCATACCCAGG
Arc	CCCCCAGCAGTGATTCATAC	CAGACATGGCCGGAAAGACT
Homer1	CACGGAGCTGGAATGTGTTA	CTGCCCCTCCAGGTCTTTAT
Grin1	AAGAATGTGACGGCTCTGCT	TGAGCTGAAGTCCGATGATG

Arc: cytoskeleton-associated protein. Homer1: homologue protein Homer1. Grin1: subunit 1 of the ionotropic glutamate NMDA receptor.

**Table 2 biomedicines-10-00588-t002:** Correlations between the extinction, measured as the difference in time spent in the naloxone-associated compartment in the extinction test minus that in the CPA test, and Arc-, GluN1-, and Homer-IR in the BLA and DG.

			r	*p*				r	*p*
**Basolateral Amygdala**	**Arc-IR**	**placebo**	−0.3327	0.5844	**Dentate Gyrus**	**Arc-IR**	**placebo**	−0.4367	0.5633
		**morphine**	−0.8039	0.1012			**morphine**	−0.8997	0.0375
		**pla + mor**	−0.3397	0.3369			**pla + mor**	−0.5254	0.1463
	**GluN1-IR**	**placebo**	−0.9397	0.0179		**GluN1-IR**	**placebo**	−0.9980	0.0020
		**morphine**	−0.6193	0.2652			**morphine**	−0.6295	0.2551
		**pla + mor**	−0.7176	0.0195			**pla+mor**	−0.7384	0.0231
	**Homer1-IR**	**placebo**	−0.2492	0.6860		**Homer1-IR**	**placebo**	0.1263	0.8737
		**morphine**	−0.9082	0.0329			**morphine**	−0.9672	0.0071
		**pla + mor**	−0.6923	0.0265			**pla + mor**	−0.7204	0.0286

**Table 3 biomedicines-10-00588-t003:** Correlations between Arc-, GluN1-, and Homer1-IR in the BLA and DG.

			GluN1-IR		Homer1-IR	
			r	*p*	r	*p*
**Basolateral Amygdala**	**Arc-IR**	**pla-CPA**	0.8853	0.0458	0.7826	0.1176
		**mor-CPA**	0.8364	0.0775	0.7997	0.1043
		**pla-ext**	0.4327	0.4668	0.8270	0.0841
		**mor-ext**	0.9581	0.0102	0.9312	0.0214
		**All experimental groups**	0.6772	0.0010	0.6803	0.0010
	**Homer1-IR**	**pla-CPA**	0.8362	0.0776		
		**mor-CPA**	0.8616	0.0605		
		**pla-ext**	0.4680	0.4267		
		**mor-ext**	0.7911	0.1109		
		**All experimental groups**	0.7505	0.0001		
**Dentate Gyrus**	**Arc-IR**	**pla-CPA**	0.9486	0.0139	0.7262	0.1647
		**mor-CPA**	0.5516	0.3351	0.7951	0.1078
		**pla-ext**	0.4522	0.5478	0.3889	0.1512
		**mor-ext**	0.8358	0.0779	0.9248	0.0245
		**All**	0.7225	0.0050	0.6187	0.0470
	**Homer1-IR**	**pla-CPA**	0.6896	0.1976		
		**mor-CPA**	0.9358	0.0193		
		**pla-ext**	−0.1689	0.8311		
		**mor-ext**	0.7839	0.1166		
		**All experimental groups**	0.5543	0.0160		

**Table 4 biomedicines-10-00588-t004:** Correlations between *Arc*, *Grin1,* and *Homer1* mRNA relative levels in the BLA and DG.

			*Grin1*		*Homer1*	
			r	*p*	r	*p*
**Basolateral Amygdala**	** *Arc* **	**pla-CPA**	0.9536	0.0119	0.9796	0.0204
		**mor-CPA**	−0.5445	0.3427	−0.1029	0.8692
		**pla-ext**	0.4700	0.4244	0.5579	0.3285
		**mor-ext**	0.1390	0.7929	0.6901	0.1972
		**All experimental groups**	0.2394	0.2959	−0.06695	0.7791
	** *Homer1* **	**pla-CPA**	0.8646	0.0586		
		**mor-CPA**	0.5222	0.2879		
		**pla-ext**	0.5789	0.3064		
		**mor-ext**	−0.06271	0.9202		
		**All experimental groups**	0.01961	0.9310		
**Dentate Gyrus**	** *Arc* **	**pla-CPA**	0.9902	0.0001	0.9117	0.0114
		**mor-CPA**	−0.3547	0.5580	0.1971	0.7506
		**pla-ext**	0.2514	0.6833	0.9902	0.0012
		**mor-ext**	0.4740	0.3422	0.5289	0.2807
		**All experimental groups**	0.1942	0.3865	0.5345	0.0104
	** *Homer1* **	**pla-CPA**	0.8715	0.0237		
		**mor-CPA**	0.5121	0.3777		
		**pla-ext**	0.3574	0.5549		
		**mor-ext**	0.9004	0.0144		
		**All experimental groups**	0.6622	0.0008		

## Data Availability

The data are available from the corresponding authors on reasonable request.

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
