# Peer review of "Molecular Mechanisms Underlying the Retrieval and Extinction of Morphine Withdrawal-Associated Memories in the Basolateral Amygdala and Dentate Gyrus"

_biomedicines, 2022, doi:10.3390/biomedicines10030588_

Round 1
Reviewer 1 Report
The work is very interesting and rich in data. The authors propose to unveil a possible biochemical marker that can be identified as the target of possible future pharmacological strategies aimed at reducing the behavior of retrieval and extinction of morphine withdrawal-associated memories. By using the conditioned place aversion (CPA) paradigm in rats we investigated some of the molecular mechanisms that occurred during the retrieval and extinction of morphine withdrawal memories in some brain areas in rats.
As mentioned, the purpose of the article is rather ambitious, but scientifically relevant, and is part of a specific line of research. The introduction is quite clear and detailed. Methods and material are adequate.
The main concern refers to the amount of data reported that are difficult to read and understand. Therefore the discussion should be very reduced and rewritten by organizing the bibliographic citations by neurotransmitter or anatomical site. Furthermore, many papers are reported, sometimes with disagreeing results; these must be selected and discussed in an understandable way.
The conclusion section is too skimpy and generic. It must contain an evaluation of the work and its possible location in the search field
Finally, the number of rats and samples examined must be reported in the tables
Author Response
The work is very interesting and rich in data. The authors propose to unveil a possible biochemical marker that can be identified as the target of possible future pharmacological strategies aimed at reducing the behavior of retrieval and extinction of morphine withdrawal-associated memories. By using the conditioned place aversion (CPA) paradigm in rats we investigated some of the molecular mechanisms that occurred during the retrieval and extinction of morphine withdrawal memories in some brain areas in rats.
As mentioned, the purpose of the article is rather ambitious, but scientifically relevant, and is part of a specific line of research. The introduction is quite clear and detailed. Methods and material are adequate.
We deeply appreciate the reviewer comments about our work.
The main concern refers to the amount of data reported that are difficult to read and understand. Therefore the discussion should be very reduced and rewritten by organizing the bibliographic citations by neurotransmitter or anatomical site. Furthermore, many papers are reported, sometimes with disagreeing results; these must be selected and discussed in an understandable way.
As reviewer suggest, the discussion has been shortened and reorganized by anatomical site and, hopefully, made more understandable and easily readable.
The conclusion section is too skimpy and generic. It must contain an evaluation of the work and its possible location in the search field
Following the reviewer instructions, the conclusions of our study have been rewritten and lengthened attempting to evaluate our research and locate it in the field.
Finally, the number of rats and samples examined must be reported in the tables
According to the reviewer’s comments, the sample size for each experimental group and experiment has been included in the figures.

Reviewer 2 Report
Manuscript ID: biomedicines-1605919-peer-review-v1
In the presented article, the authors describe Western Blot, RTqPCR and IHC data obtained in rats’ conditioned (by morphine withdrawal with naloxone) place aversion model. Described data are focused on the changes in the level of p/mTORC and its targets: Arc, GluN1 and Homer in the basolateral amygdala and dentate gyrus of hippocampus evoked by the retrieval and extinction of aversive memories. The scientific approach presented in the manuscript is interesting and the results are original.
My critical remarks concern mainly methodological elucidations and results presentation. The manuscript should be corrected and completed therefore, my recommendation for it is MAJOR REVISION.
MAIN COMMENTS:
1)The title is engrossing, however, a little confusing in the context described in the manuscript, experimental groups. I believe authors by retrieval group mean "CPA" group. Please consider highlighting in the text which group is "retrieval". It would be connected with changes in the introduction (1st paragraph- lines 37-44).
2) Please specify "drug...." in line 32. Did you mean >morphine withdrawal associated memory<
3)Is there any specific reason why the experiment was conducted on the male sex of rats only?
4) In chapter 2.3 "Behavioural procedures" authors did not indicate clearly which treatment is responsible for retrieval of morphine-withdrawal memories. It is not clear also from the scheme of experimental procedures on animals illustrated in Fig.1A. Please complete/correct.
5) Line 135. Please add a short description of all phases of the CPA protocol before a detailed description of these phases in subsections 2.3.3.1 - 2.3.3.3. You can add here reference to the scheme of the experiment presented in Fig 1A.
6) line 157-158: Please specify symptoms of the emotional withdrawal syndrome after naloxone you expected and explain if/ in which way you verified their presence.
7) Please organize and complete the information contained in chapter 2.5 "Electrophoresis and immunoblotting":
First, you should describe a procedure of Western Blot you performed and later the method of band calculation. In the current version, information is chaotic and not precise. E.g.:
-you describe the stripping procedure after the method of blot calculation. Please reorganize
-There is no information about the buffer used for protein isolation from micro punches (the reader cannot understand which fraction of protein was analysed). Also, lack is information about protease, phosphatase inhibitors used in the procedure. Please complete your description
- There is no information in which way variation in protein loading was normalized. Typically, control protein (actin, calnexin, tubulin etc) is assessed
8) In chapter 2.6. "RNA extraction..."
- please explain or correct if applicable why Trizol was used for RNA homogenization in the "column" isolation method instead of lysis buffer from Qiagen RNeasy Kit.
-please specify the amount of RNAse inhibitor used per sample in the RT procedure and complete information about the origin of RNAse inhibitor.
-please describe a method of RTqPCR data calculation.
9) In chapter 2.9, please complete the information about the number of animals per group analyzed by IHC triple labelling. Moreover, please complete the method of data quantification.
10) line 296: Please specify parameters for Pearson's correlation you performed. Please provide necessary details to understand the causal connection of the used method, enable correct assessment of the quality of performed studies and improve understanding of data presented in Tables
11)In the title of graphs and the body text of the Article you describe the effects of morphine retrieval and morphine extinction while in figures you point out "CPA" and "extinction". To avoid confusion, please consider changing the figures name "CPA" into retrieval.
12) Please improve the visibility of colocalization, e.g. in Fig 2D, 2E and Fig 3D and 3E by adding arrows in places of colocalization.
13)Please correct the numbers in the presented Tables. In the current version, we can find Table 2 and Table 4 while table 1 and table 3 are absent
14)In the Results section, please complete statistical reports, I mean add a statistical report also for data where you did not find significance.
MINOR COMMENTS:
1)I would argue with the sound of 1st sentence in the abstract. Although the addictive potential of opiates is high, more dangerous in the clinics (for pain management) is that opioids induce tolerance. It causes the necessity of increasing doses to obtain the same therapeutic effect and the risk of respiratory depression and lethality after opioids. Addictive potential in this context is not so important. Please consider rephrasing the sentence to be more specific.
2)Once introduced abbreviation should be used consequently. Because you introduced SEM abbreviation in line 293, please change "standard error of the mean" to >SEM< in the rest of the article and the figure captions.
3) In Line 148, the introduced abbreviation “nx” is not used later on and should be removed from the text.
4)Line 328, “hypothalamic DG”. Did you mean >hippocampal DG<
Author Response
In the presented article, the authors describe Western Blot, RTqPCR and IHC data obtained in rats’ conditioned (by morphine withdrawal with naloxone) place aversion model. Described data are focused on the changes in the level of p/mTORC and its targets: Arc, GluN1 and Homer in the basolateral amygdala and dentate gyrus of hippocampus evoked by the retrieval and extinction of aversive memories. The scientific approach presented in the manuscript is interesting and the results are original.
My critical remarks concern mainly methodological elucidations and results presentation. The manuscript should be corrected and completed therefore, my recommendation for it is MAJOR REVISION.
MAIN COMMENTS:
1)The title is engrossing, however, a little confusing in the context described in the manuscript, experimental groups. I believe authors by retrieval group mean "CPA" group. Please consider highlighting in the text which group is "retrieval". It would be connected with changes in the introduction (1st paragraph- lines 37-44).
We deeply appreciate the suggestions of the reviewer. We have slightly modified the title of the manuscript, as suggested.
On the other hand, we have not used the word “retrieval” for the CPA group because in the extinction test animals also retrieved memories, although those were “extinction memories” and not the ones associated with morphine withdrawal. Nevertheless, and following the reviewer’s recommendation, we have clarified which group corresponds with the retrieval of morphine withdrawal-paired memories in the manuscript (page, 3, section 2.3.3, lines, 142-145) and in Fig. 1A.
2) Please specify "drug...." in line 32. Did you mean >morphine withdrawal associated memory<
According to reviewer suggestion, we have replaced “drug-associated aversive memories” with “morphine withdrawal-associated aversive memories”.
3) Is there any specific reason why the experiment was conducted on the male sex of rats only?
There are not specific reasons for using only males other than back when the funding was granted for this investigation the research was only planed in male animals. However, due to the misrepresentation of female sex in preclinical and clinical biomedical research, male and female animals have been included in all the posterior research projects that have been granted to our group.
4) In chapter 2.3 "Behavioural procedures" authors did not indicate clearly which treatment is responsible for retrieval of morphine-withdrawal memories. It is not clear also from the scheme of experimental procedures on animals illustrated in Fig.1A. Please complete/correct.
Following reviewer’s recommendation, we have clarified which group corresponds with the retrieval of morphine withdrawal-paired memories in the text (page 3, section 2.3.3, lines 142-145) and in Fig. 1A.
5) Line 135. Please add a short description of all phases of the CPA protocol before a detailed description of these phases in subsections 2.3.3.1 - 2.3.3.3. You can add here reference to the scheme of the experiment presented in Fig 1A.
As recommended by the reviewer, we have added a short paragraph that includes all the phases of the CPA protocol before their detailed description (page 3, section 2.3.3, lines 137-145) and we have referenced it to the scheme of the experiments presented in Fig 1A.
6) line 157-158: Please specify symptoms of the emotional withdrawal syndrome after naloxone you expected and explain if/ in which way you verified their presence.
We administered a low dose of naloxone (15 μg/kg) in order to minimize the physical signs of morphine withdrawal, given that is known that the aversion that animals develop to the withdrawal-associated environment is due to the emotionally aversive state of the abstinence syndrome but not to its physical symptoms (Schulteis, G et al. J. Pharmacol. Exp. Ther. 1994, 271, 1391-1398; Myers, K.M. et al. Nat. Protoc. 2012, 7, 517-526; Myers, K.M. & Carlezon, W.A., Jr. Biol. Psychiatry 2010, 67, 85-87). The only symptom that we expected of this emotionally negative state was the aversion to the withdrawal-associated compartment shown by the opiate-dependent animals in the CPA test. However, we did not detect, for instance, one of the most characteristic physical symptoms of morphine withdrawal such as weight loss: the weight loss of morphine dependent rats 1 h after naloxone administration was not statistically different from that observed one hour after saline injection (data not reported).
We have rephrased the last sentence of the section 2.2 (page 3, lines 111-114) and added an explanation in the results 3.1 section (page 8, lines 327-329) to explain how we verified the existence of an emotional withdrawal syndrome in morphine-dependent rats.
7) Please organize and complete the information contained in chapter 2.5 "Electrophoresis and immunoblotting":
First, you should describe a procedure of Western Blot you performed and later the method of band calculation. In the current version, information is chaotic and not precise. E.g.:
-you describe the stripping procedure after the method of blot calculation. Please reorganize
In accordance with the reviewer comment, we have detailed the stripping procedure just after the detection of immunoreactivity.
-There is no information about the buffer used for protein isolation from micro punches (the reader cannot understand which fraction of protein was analysed). Also, lack is information about protease, phosphatase inhibitors used in the procedure. Please complete your description
Because the nuclear fraction of these samples was used to evaluate some histones alterations, for their processing we followed the protocol publish by Beldjoud et al, 2016 (Beldjoud H, et al. Curr Protoc Neurosci. 2016; 76:4.38.1-4.38.20), and we performed the WB and RT-qPCR analyses presented in this study in the cytoplasmic fraction. We have added this information and the above reference in the new version of the manuscript (page, 5, section 2.5, lines 213-214).
- There is no information in which way variation in protein loading was normalized. Typically, control protein (actin, calnexin, tubulin etc) is assessed.
As in this study we report the phosphorylation ratio of mTOR (pmTOR/mTOR) but not the levels of pmTOR nor mTOR, the influence of the variations in protein loading is annulled. Nevertheless, on the one hand we determined protein concentration in each sample before western blot analyses and loaded 15 μg of each sample and, on the other hand, after mTOR and pmTOR determinations we stripped the PVDF membranes and incubated them with an antibody against glyceraldehyde-3-phosphate dehydrogenase (GAPDH).
8) In chapter 2.6. "RNA extraction..."
- please explain or correct if applicable why Trizol was used for RNA homogenization in the "column" isolation method instead of lysis buffer from Qiagen RNeasy Kit.
The RNA homogenization vas performed with Quiazol, provided by Qiagen. We have corrected this data in the manuscript (page 6, line 234).
-please specify the amount of RNAse inhibitor used per sample in the RT procedure and complete information about the origin of RNAse inhibitor.
The following sentence has been added to the section 2.6 of the manuscript (page 6, lines 239-240): To avoid RNA degradation, RNAase inhibitors (Applied Biosystems) were used at a final concentration of 1.0 U/μL.
-please describe a method of RTqPCR data calculation.
For the RT-qPCR, amplifications were carried out in triplicate and the relative expression of target genes was determined by the ΔΔCT method. We have included this information in page 6, section 2.6, lines 244-245.
9) In chapter 2.9, please complete the information about the number of animals per group analyzed by IHC triple labelling. Moreover, please complete the method of data quantification.
As the reviewer indicated, we have added the number of animals per experimental group (n = 4 - 5) in the triple immunolabelling (page 7, line 301). The details of the quantification method have also been stated (section 2.9, lines 298-302) as follows: “Time exposure (2 s) and settings for both nuclei were constant through experimental groups, and images were captured by a blinded investigator at 20X magnification. Quantification of the images was performed by using FIJI software v. 2.1.0/1.53c (NIH ImageJ, Bethesda, MD, USA). Firstly, “.lif” documents exported from LAS X were opened as “hyperstack” through the Bioformat plugin; then, the region of interest corresponding to BLA and DG was selected manually in one of the channels and the same region was replicated automatically in the following captured channels. Afterwards, the mean grey value of these regions was measured automatically by this software. Three to six sections of each animal (n = 4-5 animals per group) were evaluated and a mean value for each animal was then calculated.”
10) line 296: Please specify parameters for Pearson's correlation you performed. Please provide necessary details to understand the causal connection of the used method, enable correct assessment of the quality of performed studies and improve understanding of data presented in Tables.
All our variables were continuous and paired. Before computing the Pearson correlation coefficient, we confirmed the absence of outliers (ROUT Method; Q = 1%) and performed several normality and lognormality tests (Anderson-Darling, D'Agostino & Pearson, Shapiro-Wilk and Kolmogorov-Smirnov tests), that were passed by all the variables. Thus, given the Gaussian distribution of our data, we computed the Pearson correlation and not the Spearman’s. The P value was two-tailed and the Confidence Interval 95%. Most of this information has been included in the section 2.10 (page 8, lines 312-318).
Because the Pearson correlation coefficient is a measure of the strength of a linear association between two variables but does not prove that a change in one variable causes a change in the other, theoretically it cannot be established a causal connection between them. Nevertheless, given the strength of the correlations detected and the abundant literature that support these postulates, we strongly believe that there is causality between the correlated variables.
11) In the title of graphs and the body text of the Article you describe the effects of morphine retrieval and morphine extinction while in figures you point out "CPA" and "extinction". To avoid confusion, please consider changing the figures name "CPA" into retrieval.
As stated earlier, we deeply appreciate the suggestion of the reviewer. We have not used the word “retrieval” for the CPA group because in the extinction test animals also retrieved memories, although those were “extinction memories” and not the ones associated with morphine withdrawal. Nevertheless, and following reviewer’s recommendation, we have clarified which group corresponds with the retrieval of morphine withdrawal-paired memories in the manuscript (page, 3, section 2.3.3, lines, 142-145) and in Fig. 1A.
12) Please improve the visibility of colocalization, e.g. in Fig 2D, 2E and Fig 3D and 3E by adding arrows in places of colocalization.
Accordingly with the reviewer suggestion, we have added arrows to the figures 2D, 2E, 3D and 3E pointing at the colocalization places.
13) Please correct the numbers in the presented Tables. In the current version, we can find Table 2 and Table 4 while table 1 and table 3 are absent
In the version of the manuscript that we uploaded for its evaluation the Tables 1, 2, 3 and 4 were present, and we have also found them in the new version that the Journal has arranged for us to modify according to the reviewers’ comments. We do not know why the file that the reviewer downloaded for revision did not include the tables 1 and 3.
14) In the Results section, please complete statistical reports, I mean add a statistical report also for data where you did not find significance.
As the reviewer recommends, we have included the statistical data for the non-significant analyses along the Results section of the manuscript.
MINOR COMMENTS:
1) I would argue with the sound of 1st sentence in the abstract. Although the addictive potential of opiates is high, more dangerous in the clinics (for pain management) is that opioids induce tolerance. It causes the necessity of increasing doses to obtain the same therapeutic effect and the risk of respiratory depression and lethality after opioids. Addictive potential in this context is not so important. Please consider rephrasing the sentence to be more specific.
In accordance with the reviewer comment, the first sentence of the abstract has been rephrased to specify that the addictive potential of opioids is not the most important deterrent factor for their prescription for pain management.
2) Once introduced abbreviation should be used consequently. Because you introduced SEM abbreviation in line 293, please change "standard error of the mean" to >SEM< in the rest of the article and the figure captions.
After the introduction of the abbreviation, we have replaced “standard error of the mean" by “SEM” throughout the manuscript.
3) In Line 148, the introduced abbreviation “nx” is not used later on and should be removed from the text.
As the reviewer suggests, the abbreviation nx has been removed from the text.
4) Line 328, “hypothalamic DG”. Did you mean >hippocampal DG<
We appreciate the reviewer observation. We have replaced “hypothalamic” by “hippocampal” DG.

Round 2
Reviewer 2 Report
The Article revised by the authors is significantly corrected and improved. The authors applied suggested corrections efficiently: improved group description, methodology, visibility of presented data, updated statistical reports. In the reviewer's opinion, the current version of the Article is much easier to read and understand. Also, the authors explained the role of the Pearson correlation coefficient measurement to link biochemical changes to withdrawal behaviour or brain structure.
I have only one minor comment.
The authors explained that they measured the phosphorylation ratio of mTOR (pmTOR/mTOR), loaded an equal amount of protein per well, assessed GAPDH level in the body text. However, it is not enough to prove that the amount of protein was the same in the well. Please add an appropriate explanation in the body text of the manuscript about possible changes in mTOR and GAPDH levels among treatment groups.
Author Response
The Article revised by the authors is significantly corrected and improved.
The authors appliedsuggested corrections efficiently: improved group description, methodology, visibility of presented data, updated statistical reports. In the reviewer's opinion, the current version of the Article is much easier to read and understand. Also, the authors explained the role of the Pearson correlation coefficient measurement to link biochemical changes to withdrawalbehaviour or brain structure.
Authors deeply appreciate the reviewer’s evaluation of our revised manuscript.
I have only one minor comment.
The authors explained that they measured the phosphorylation ratio of mTOR (pmTOR/mTOR), loaded an equal amount of protein per well, assessed GAPDH level in the body text. However, it is not enough to prove that the amount of protein was the same in the well. Please add an appropriate explanation in the body text of the manuscript about possible changes in mTOR and GAPDH levels among treatment groups.
Following reviewer’s suggestion, we have clarified that the WB data are an index of the mTOR phosphorylation ratio but not a marker of mTOR nor p-mTOR levels in the result section (page 9, lines 356-357).
